# Spatial suppression promotes rapid figure-ground segmentation of moving objects

Duje Tadin[1,2], Woon Ju Park [1,3], Kevin C. Dieter [1], Michael D. Melnick[1], Joseph S. Lappin[4] & Randolph Blake [4]

Segregation of objects from their backgrounds is a fundamental visual function and one that is particularly effective when objects are in motion. Theoretically, suppressive center-surround mechanisms are well suited for accomplishing motion segregation. This long-standing hypothesis, however, has received limited empirical support. We report converging correlational and causal evidence that spatial suppression of background motion signals is critical for rapid segmentation of moving objects. Motion segregation ability is strongly predicted by both individual and stimulus-driven variations in spatial suppression strength. Moreover, aging-related superiority in perceiving background motion is associated with profound impairments in motion segregation. This segregation deficit is alleviated via perceptual learning, but only when motion segregation training also causes decreased sensitivity to background motion. We argue that perceptual insensitivity to large moving stimuli effectively implements background subtraction, which, in turn, enhances the visibility of moving objects and accounts for the observed link between spatial suppression and motion segregation.

[1] Department of Brain & Cognitive Sciences; Center for Visual Science, University of Rochester, Rochester, NY 14627, USA. [2] Departments of Ophthalmology and Neuroscience, University of Rochester Medical Center, Rochester, NY 14642, USA. [3] Department of Psychology, University of Washington, Seattle, WA 98195, USA. [4] Department of Psychology, Vanderbilt University, Nashville, TN 37240, USA. Correspondence and requests for materials should be addressed to D.T. (email: dtadin@ur.rochester.edu)

Segregation of objects from their backgrounds is one of vision's most important tasks[1]. This essential step in visual processing, termed figure-ground segmentation, has fascinated neuroscientists and psychologists since the early days of Gestalt psychology. Visual motion is an especially rich source of information for rapid, effective object segregation[1–5]. A stealthy animal cloaked by camouflage immediately loses its invisibility once it begins moving, just as does a friend you're trying to spot, waving her arms amongst a bustling crowd at the arrival terminal of an airport. While seemingly effortless, visual segregation of moving objects invokes a challenging problem that is ubiquitous across sensory and cognitive domains: balancing competing demands between processes that discriminate and those that integrate and generalize[6–11]. Figure-ground segmentation of moving objects, by definition, requires highlighting of local variations in velocity signals. This, however, is in conflict with integrative processes necessitated by local motion signals that are often noisy and/or ambiguous. Achieving an appropriate and adaptive balance between these two competing demands is a key requirement for efficient segregation of moving objects[6,12].

How does our visual system accomplish this critical task of segregating objects in motion? A longstanding hypothesis postulates a key role of antagonistic center-surround mechanisms[13]. Neurons with such properties are found in motion selective brain regions, most prominently in cortical area MT, a key area for motion processing[14]. A typical center-surround neuron responds strongly when its receptive field center is stimulated with motion in its preferred direction. However, when the spatial extent of stimulation is enlarged to include the receptive field surround, the neuron's response is suppressed[14,15]. Theoretical work shows that this sensitivity to small moving stimuli paired with insensitivity to wide-field, uniform motions is well suited to support segmentation of moving figures from their backgrounds[13,16–18]. However, direct behavioral evidence for this link is lacking, and even indirect neurophysiological evidence is limited[12,19–21]. In fact, motion segregation could be accomplished by specialized processes not involving center-surround mechanisms[22], with center-surround mechanisms playing a role in other visual processes, including redundancy reduction, input normalization, estimation of optic flow, heading direction, 3D shape-from-motion, and detection of edge discontinuities[23–30]. In sum, whether motion segregation ability indeed depends on suppressive center-surround mechanisms is an open empirical question.

To address this question, we exploited behavioral correlates of surround suppression[12], focusing on strong suppressive effects that are prominent in direction discriminations of brief moving stimuli. As the size of such stimuli increases, observers experience progressively increasing difficulty at perceiving visual motion[31]. This behavioral phenomenon, termed spatial suppression (to differentiate it from neural surround suppression mechanisms), is thought to reflect surround suppression mechanisms within cortical area MT[32–35]. A defining property of behavioral spatial suppression is that it adapts to stimulus characteristics; spatial suppression is strong for motion stimuli that are highly visible and weak for those that are either noisy or low in contrast[31]. This adaptability of spatial suppression parallels theoretical constraints associated with competing demands between segregation and integration of moving signals[6]. Functional roles of spatial suppression, however, remain unclear and, indeed, at face value, it appears to be detrimental to our ability to perceive visual motion. Here, we test the hypothesis that mechanisms of spatial suppression directly enable rapid segregation of moving objects by suppressing motion signals belonging to the background. We propose that spatial suppression, in essence, functions equivalently to background subtraction algorithms used in computer vision—algorithms that are particularly effective when the

background can be reliably identified as is the case with moving backgrounds[17,36–39]. Within this framework, in other words, large, uniform moving stimuli are treated as backgrounds owing to spatial suppression. This is consistent with classic observations by Rubin and others[40,41] that smaller and larger stimuli are more likely to be perceived as figures and backgrounds, respectively.

The results provide converging correlational and causal evidence for the hypothesis that spatial suppression of background motion signals is critical for rapid segmentation of moving objects. We first show that stimulus conditions associated with strong spatial suppression (high contrast and fast speed) also support efficient motion segregation. Moreover, individuals that exhibit strong spatial suppression also excel at motion segregation. This correlational relationship holds for both neurotypical younger adults and for older adults who show age-related weakening of spatial suppression[42]. Finally, we show that age-related weakening of spatial suppression[42] can be reversed as a result of motion segregation training. Taken together, these results provide strong converging evidence for a functional linkage between spatial suppression and motion segregation.

## Results

**Overview of the results**. The results are organized into four sections. The first two sections provide indirect, correlational evidence for a link between spatial suppression and motion segregation. First, by manipulating stimulus characteristics, we show that humans excel at motion segregation under stimulus conditions associated with strong spatial suppression (Exp. 1 and 2). Here, segregation of moving objects from a moving background is remarkably effective, matching and even surpassing detection of moving objects seen against static backgrounds. Next (Exp. 3), by using an individual differences approach, we show that individual variations in spatial suppression strength correlate with observers' ability to segregate moving objects. Motivated by weakening of spatial suppression in senescence[42] we also tested older adults and found that age-related improvements in the relative visibility of large, background motions were associated with pronounced deficits in motion segregation.

The third section tests a prediction derived from correlational experiments in the first two sections: if human motion segregation ability is largely derived from spatial suppression mechanisms, then an intervention that improves motion segregation should also cause corresponding changes in spatial suppression (Exp. 4). Results confirmed this prediction by showing that improvements in motion segregation produced by perceptual learning are accompanied by a corresponding strengthening of spatial suppression. The fourth section describes a model that aims to gain insights into what mechanistic changes might underlie the observed change in spatial suppression. Depending on the reader's preferences, the results can be either read in its entirety (as gradually building toward more conclusive experiments) or a reader can skip to the third section ("Causal linking of motion segregation and spatial suppression") treating the first two sections similar to Supplementary materials.

**Covariation of motion segregation and spatial suppression**. We start by testing a straightforward prediction. If spatial suppression underlies visual motion segregation, then stimulus conditions that strengthen spatial suppression should aid motion segregation and vice versa (Exp. 1). In other words, a motion-defined object should be more visible when its background motion is less visible. To test this hypothesis, we asked observers to (a) identify a motion-defined object imbedded in a large moving background pattern (motion segregation task; Fig. 1a), and (b) identify motion direction of the background motion presented on its own (motion discrimination

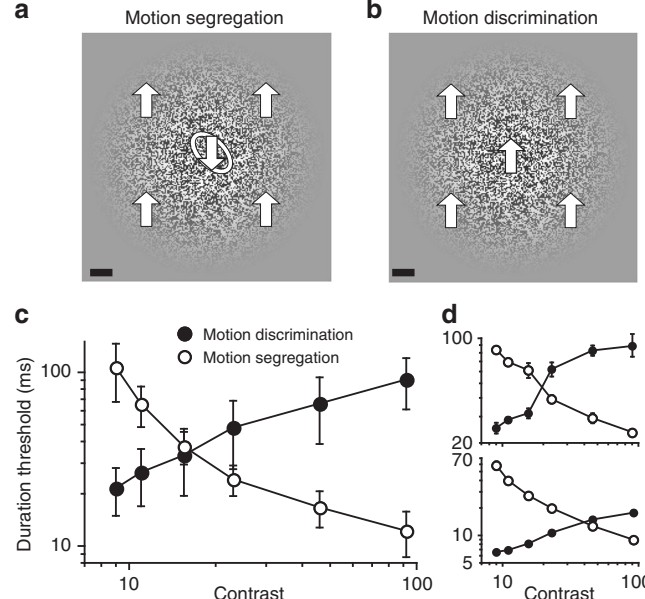

**Fig. 1** Motion segregation is effective when background motion is less discriminable. **a** In the motion segregation task, observers identified the tilt of an oval motion-defined object presented on a moving background. The object could be either tilted left (as shown) or tilted right. Block arrows and the object outline are shown for illustration purposes only. Scale bar is 1°. **b** In the motion discrimination task, the background motion from panel A was presented in isolation, and observers were simply asked to identify its motion direction (which could be either upward or downward). Scale bar is 1°. **c** Group data showing the opposite effects of stimulus contrast on motion discriminations and motion segregation. **d** Data for two individual observers. All error bars are s.e.m.

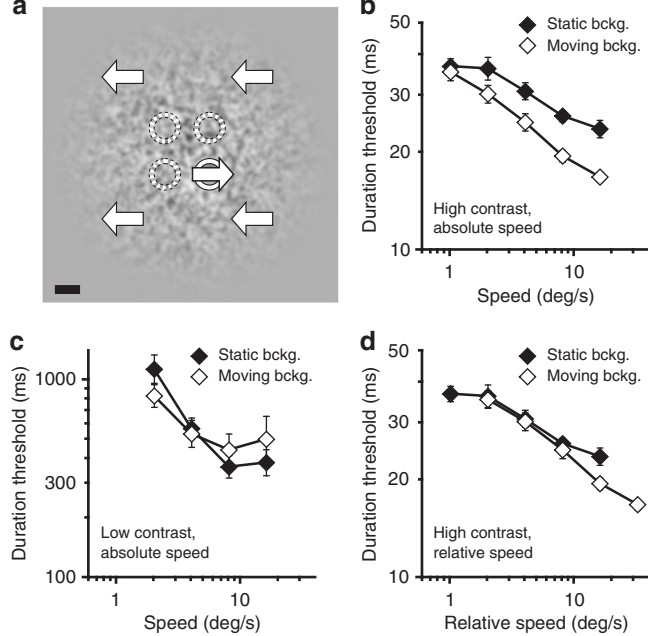

**Fig. 2** Motion segregation on moving and static backgrounds. **a** Observers identified the location of a moving target which could occur in one of four possible locations (shown with dotted circles for illustration purposes only). The target background was either moving (as illustrated) or stationary. Scale bar is 2°. **b** Effects of speed on task performance for moving and static backgrounds at high contrast. **c** Same as (**b**), but at low contrast. **d** Same data as in (**b**), but with speed expressed as relative to the background. Error bars are s.e.m.

task; Fig. 1b). Both tasks were done across a range of contrasts. As expected[31], motion discriminations of the large background motion considerably worsened with increasing contrast (Fig. 1c), a result consistent with spatial suppression at high contrasts (note that because of a blurred spatial envelope (Fig. 1a, b) increasing contrast also results in modest increases in the effective stimulus size, which also strengthens spatial suppression[31]). On the other hand, motion segregation gradually improved with increasing contrast (Fig. 1c), exhibiting a strong negative correlation with decreases in the visibility of the background motion (Fig. 1c, d; group result: Pearson $r = -0.977$, median correlation across observers: $-0.983$). Notably, at the maximum contrast tested, thresholds for simply perceiving background motion direction were over 7 times higher than thresholds for segregating moving objects on the same size background.

Next, we further examined the effectiveness of motion segregation. Here, we contrasted motion segregation—the ability to segment moving objects on moving backgrounds—with detection of moving objects on static backgrounds, an ostensibly much simpler task (Exp. 2). The observers were asked to identify the location of a moving object, which could appear in one of four different positions (Fig. 2a). The key manipulation was the nature of the background: it was either stationary or moving. At high contrast, observers identified an object's location more quickly when the background was also moving (Fig. 2b, $F_{1,5} = 72.27$, $P < 10^{-3}$, $\eta^2 = 0.935$). The advantage of the moving background, however, weakened both at slow speeds (Fig. 2b; interaction: $F_{4,20} = 4.57$, $P = 0.009$, $\eta^2 = 0.478$) and at low contrast (Fig. 2c, $F_{1,5} = 0.063$, $P = 0.81$, $\eta^2 = 0.012$). This pattern of results is consistent with speed- and contrast-dependency of spatial suppression; suppression weakens both at low contrast and

at slow speeds[31,43]. Thus, it appears that the visual system excels at motion segregation under conditions that promote insensitivity to background motion – a result consistent with human vision subtracting background motion to accomplish motion segregation.

How effective is this putative background subtraction? To answer that question, we expressed the results in Fig. 2b as a function of the relative speed between the object and the background (i.e., fully correcting for the larger object-background displacements in the moving background condition). Plotted in that way, results (Fig. 2d) show that the presence of the moving background has no negative effect on motion segregation. In fact, we only found evidence for a trend for better motion segregation performance with moving backgrounds ($F_{1,5} = 6.51$, $P = .051$, $\eta^2 = 0.566$). In other words, motion segregation with moving backgrounds can function at least as efficiently as detection of single moving object on a static background.

To summarize results thus far, we show that the visual system exhibits remarkable effectiveness at differentiating figural regions from their background based on motion under stimulus conditions that also promote strong spatial suppression. In such cases, segregation of moving objects can be several times more efficient than the perception of background motion per se. In the following experiments, we aim to determine whether perceptual insensitivity to background motion, in fact, causes increased visibility of object motion.

**Explaining individual differences in motion segregation.** Key insights into visual mechanisms can be derived from individual differences[44–46]. In context of the proposed functional link between spatial suppression and motion segregation, the hypothesis is that, for both young and older adults, spatial suppression strength should predict their ability to segregate moving

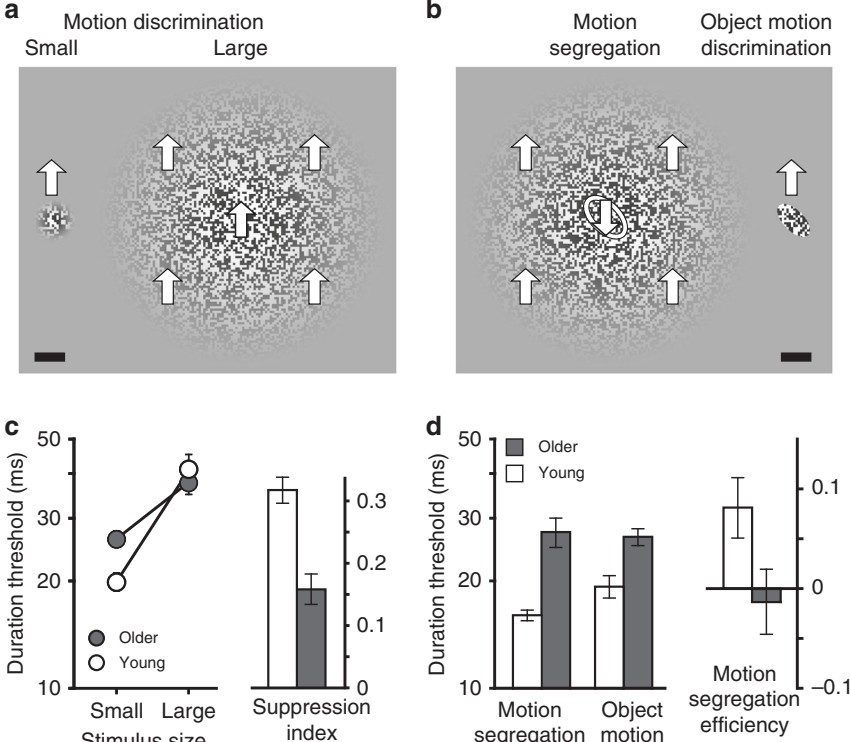

**Fig. 3** Age-related changes in spatial suppression and motion segregation. In separate blocks, observers discriminated motion directions (up vs. down) of small and large stimuli (**a**), and that of an oval object (**b**, right). In the motion segregation task (**b**, left), observers identified the orientation of a motion-defined object (left tilt vs. right tilt). Block arrows and the object outline are shown for illustration purposes only. Scale bar is 1°. (**c**) The effect of stimulus size on motion direction discriminations in young and older adults. Suppression index is defined as $\log_{10}$(large size threshold) – $\log_{10}$(small size threshold). (**d**) Motion segregation and motion discrimination of an oval shape in young and older adults. Motion segregation efficiency is defined as $\log_{10}$(figure-motion threshold) – $\log_{10}$(motion segregation threshold). All error bars are s.e.m.

objects (Exp. 3). We included older adults because senescence is associated with atypically weak spatial suppression[42], raising a question of whether this atypicality is also associated with impairments in motion segregation.

To answer this question, we used four different tasks. The first two tasks (Fig. 3a) involved motion direction discriminations (upward vs. downward) of either a small (Fig. 3a, left) or a large moving stimulus (Fig. 3a, right). By testing how much performance deteriorates with increasing stimulus size, we were able to assess spatial suppression strength. The third task assessed motion segregation ability (as in Exp. 1), with observers asked to identify the tilt of a motion-defined object (Fig. 3b, left). Finally, to estimate the visibility of the target object motion on its own, we measured observers' ability to discriminate the target object motion when shown without the moving background (Fig. 3b, right).

Starting with consideration of motion discrimination (Fig. 3a), we found that both young and older observers exhibited increased thresholds for perceiving direction of motion with increasing size (Fig. 3c left, $F_{1,36} = 223$, $P < 10^{-8}$, $\eta^2 = 0.861$), a result indicating spatial suppression. Consistent with prior studies[42,47], older adults were less affected by the increasing stimulus size (Fig. 3c left, $F_{1,36} = 25$, $P = 10^{-5}$, $\eta^2 = 0.405$). To quantify suppression strength, we computed the "suppression index" for each participant—a measure indexing one's insensitivity to motion direction of large stimuli relative to the person's ability to perceive motion of small stimuli[42,48]. The suppression index was considerably weaker in older adults (Fig. 3c right, $t_36 = 4.97$, $P = 10^{-5}$, Cohen's $d = 1.61$).

Turning to motion segregation (Fig. 3b), young adults were able to perceive motion-defined objects with ease; their motion segregation thresholds were actually better than their ability to

discriminate target object motion when shown without its moving background (Fig. 3d left, $t_{36} = 2.69$, $P = 0.01$, Cohen's $d = 0.76$). Notably, however, older adults exhibited highly impaired motion segregation compared to younger adults (Fig. 3d left, $t_{36} = 5.8$, $P = 10^{-6}$, Cohen's $d = 1.84$). We can then estimate "motion segregation efficiency" by indexing motion segregation thresholds relative to the observers' ability to discriminate motion of the target object motion when shown in isolation. Note that this index accounts for older observers' decreased ability to perceive motion of the target object motion in isolation. Even with this correction, older adults' motion segregation efficiency was lower than that of young observers (Fig. 3d right, $t_{36} = 2.12$, $P = 0.041$, Cohen's $d = 0.69$). In sum, we found weaker spatial suppression and impaired motion segregation in older adults. The key question is whether these two processes are functionally related. We consider this question next, first with a correlational approach and then, in the subsequent section, with a causal approach.

With one telling exception, correlations among four experimental conditions (Fig. 3a, b) were strongly positive ($0.55 < r < 0.90$; all $P < 10^{-4}$). This simply indicates that observers who performed well on one motion perception task tended to perform well on other motion tasks. The notable exception was a lack of significant relationship between motion segregation thresholds and thresholds for discriminating large background motions ($r = -0.07$; $P = 0.67$). This result suggests that in addition to general motion sensitivity that should affect both tasks similarly, there is at least one other factor that has an opposite effect. To investigate this possibility, we statistically controlled for the general motion sensitivity by computing correlations between motion segregation and discriminations of both small and large stimuli, while controlling for large and small stimulus thresholds, respectively.

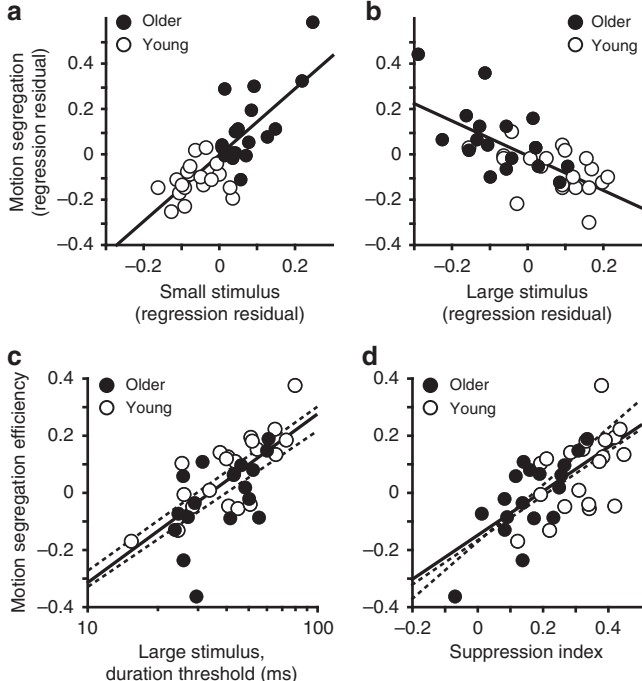

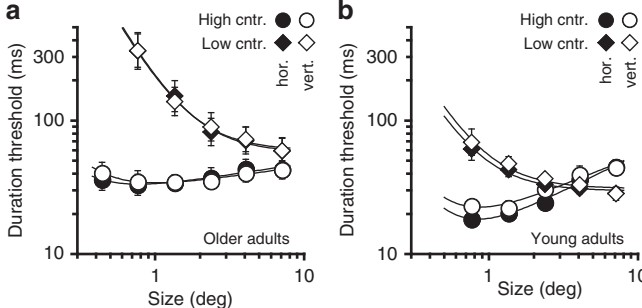

**Fig. 5** Effects of size and contrast on spatial summation and surround suppression in older adults. **a** Pre-training data for older adults. The results revealed spatial summation at low contrast and weak spatial suppression at high contrast, with no differences between horizontal and vertical motions. Data were fit with the model described in the "Methods" section. **b** Data for young adults shown for comparison purposes. All error bars are s.e.m.

**Fig. 4** The relationship between motion segregation and spatial suppression. **a** The relationship between standardized residuals for motion segregation and small stimulus tasks after regressing thresholds for the large stimulus from both. In both (**a**) and (**b**), higher values indicate worse performance. **b** The relationship between standardized residuals for motion segregation and large stimulus tasks after regressing thresholds for the small stimulus from both. Motion segregation efficiency is predicted by both **c** poor motion discriminations of large stimuli and **d** strong spatial suppression. Here, higher values indicate better motion segregation efficiency. Dashed lines are linear fits to young and older adult data. Solid lines are linear fits to the pooled data

These partial correlations revealed that the link between motion perception and motion segregation strongly depended on stimulus size. Good motion segregation performance was associated with low thresholds for small stimuli (Fig. 4a; $r = 0.78$, $P = 10^{-8}$) but also with inability to perceive large, background-like motions (Fig. 4b; $r = -0.68$, $P = 10^{-5}$). The same pattern of results was observed in a complementary multiple regression analysis ($R^2 = 0.62$, $F_{2,35} = 28.2$, $P = 10^{-7}$), where thresholds for small moving stimuli had a significant positive contribution ($B = 1.43$, $t_{35} = 7.48$, $P = 10^{-8}$), while large stimulus thresholds had a significant negative contribution ($B = -0.77$, $t_{35} = -5.35$, $P = 10^{-5}$) to motion segregation performance. These results support the hypothesis that inability to perceive large moving stimuli at brief durations may underlie motion segregation ability. Indeed, thresholds for discriminating large moving stimuli were correlated with motion segregation efficiency, such that observers who were slower (i.e., needed longer stimulus durations) at perceiving large moving stimuli had better motion segregation efficiency (Fig. 4c; $r = 0.69$, $P = 10^{-6}$). Similarly, suppression strength predicted motion segregation efficiency (Fig. 4d; $r = 0.68$, $P = 10^{-6}$). The same pattern of results was found when we separately considered older and younger observers (Fig. 3c, d, dashed lines). In sum, we find strong correlational evidence for a link between spatial suppression and motion segregation.

**Causal linking of motion segregation and spatial suppression.** The first two sections show that spatial suppression correlates

with motion segregation, both within and between observers. If these correlational links indeed reflect a functional link between motion segregation and mechanisms underlying spatial suppression, then the following prediction should hold: causing improvements in motion segregation should also come with concomitant changes in spatial suppression. To test this prediction (Exp. 4), we utilized perceptual learning to produce motion segregation improvements, hypothesizing that these improvements will result in the strengthening of spatial suppression (i.e., worse perception of large background-like motions). This approach is similar to prior studies that have exploited perceptual learning to forge causal links[49,50].

We focused on older adults who, because of abnormally weak spatial suppression[42,47] and poor motion segregation (Fig. 3c, d), have more room for improvements. In contrast, remarkably efficient motion segregation in young adults (Figs. 1–3) should limit or even preclude further training-dependent improvements (we confirmed this in our pilot explorations). Additionally, by focusing on older adults, we were able to determine whether perceptual training can at least partially alleviate age-related deficits in motion segregation (Fig. 3d).

During pre-training sessions, we measured size-dependency of motion discrimination thresholds for both horizontal and vertical motions at low (7%) and at high (99%) contrast. We included low contrast motions to test whether changes induced by motion segregation training are specific to suppressive processes, processes that are restricted to higher contrasts[31,51] or simply result in more general motion perception improvements. The pre-training results for older adults (Fig. 5a) revealed spatial summation at low contrast and relatively weak or non-existent spatial suppression at high contrast, replicating earlier findings[42]. These data were fit with a simple descriptive model to characterize the strengths of summation and suppression[47] (Equation 1, see "Methods" section). For comparison, we performed the same measurements in young adults (Fig. 5b). At high contrast, in addition to generally lower thresholds for young adults, the two groups differed in suppression strength, as indicated by a higher suppression slope ($k_2$) in young adults (0.50 vs. 0.17, $F_{1,7} = 26$, $P = 0.001$, $\eta^2 = 0.785$), replicating group differences in spatial suppression shown in Fig. 3. For both groups, there was neither a main effect of motion direction nor an interaction (all $P > 0.11$). At low contrast, consistent with prior work[42], both groups had similarly low suppression (suppression slope: 0.067 vs. 0.096, $F_{1,7} = 0$, $P = 1$), with older adults exhibiting a pronounced elevation in thresholds. These data also help rule out size-dependent differences in attention as causing the observed age-related changes in motion perception. In this

alternative explanation, differences in motion discriminations (Fig. 3c) and poor motion segregation (Fig. 3d) could be attributed to older adults being better at attending to large compared to small moving stimuli. However, in that case, we would expect similar results at low and at high contrast.

Next, older adults underwent 16 sessions of perceptual training on a motion segregation task (as shown in the left side of Fig. 3b, but either with horizontal or vertical motions). By limiting training to one direction of motion, we were able to test the hypothesis, derived from evidence that perceptual learning for motion is directionally selective[52,53], that changes in spatial suppression will be specific to that trained direction. Resultant thresholds were fit with an exponential function, whose slope provides an index of the magnitude of improvement, i.e., the degree of learning. The slope was negative (Fig. 6a; $-0.054 \pm 0.02$ s.e.m.), indicating that, on average, motion segregation thresholds decreased over time. Different observers, however, exhibited varying amounts of learning, ranging from no learning to a ~25% improvement in motion segregation (abscissa in Fig. 6c). Later in this paper, we will return to these individual differences.

A comparison of pre and post-training direction discriminations (Fig. 6b, gray vs. black lines) reveals that motion segregation training affected observers' ability to perceive high contrast moving stimuli. Namely, spatial suppression strength for the trained motion directions increased after training, as evident by a relative increase in thresholds for large, high contrast motions (Fig. 6b, i.e., more pronounced upward bend for larger stimulus sizes but only for the trained motion direction). Analyzing results from individual observers, we found significant suppression slope increases in 3/5 observers for the trained motion direction (all $P < 0.046$; no changes for the untrained motion direction, all $P > 0.21$; see "Methods" section for bootstrap analysis details). Crucially, whether an observer showed a change in suppression strength was predicted by the amount of perceptual learning during the training phase; individual learning slopes strongly correlated with post-training increases in suppression strength (Fig. 6c; $r = 0.97$, $P = 0.007$). For the observers who showed significant perceptual learning, training-induced changes in spatial suppression were substantial (Fig. 6d). For the largest stimulus size, post-training thresholds were almost twice as high for the trained as the untrained motion directions, with the suppression slope more similar to young adults (Fig. 6d; 0.37 vs. 0.50; although thresholds for older adults remained atypically elevated). Post-training suppression slopes for observers who exhibited statistically significant learning were significantly higher than both their pre-training, trained-direction suppression slopes ($t_2 = 8.1$, $P = 0.015$, Cohen's $d = 1.37$) and post-training, untrained-direction suppression slopes ($t_2 = 6.9$, $P = 0.02$, Cohen's $d = 1.27$).

Notably, post-training changes were restricted to large stimulus sizes. We found no changes in observers' ability to see small moving stimuli even though those stimuli were similar in size to the moving shape discriminated during training (gray bars in Fig. 6b, d). This size-specificity provides further evidence that reduced sensitivity to background motion is what drives improvements in motion segregation. Turning to low contrast data, we found very little effect of motion segregation training (Fig. 6e). We observed no evidence for spatial suppression either before or after training, with 19/20 95% CI around suppression slope ($k_2$) including 0 (5 observers × 2 motion directions, both pre and post-training). Evidently, motion segregation training has no significant effect on motion discriminations at low contrast (Fig. 6e) and small stimulus size (Fig. 6b, d)—stimulus conditions that are unaffected by spatial suppression[31]. Moreover, the observation that training did not affect the ability to perceive small moving stimuli of any contrast (while improving motion

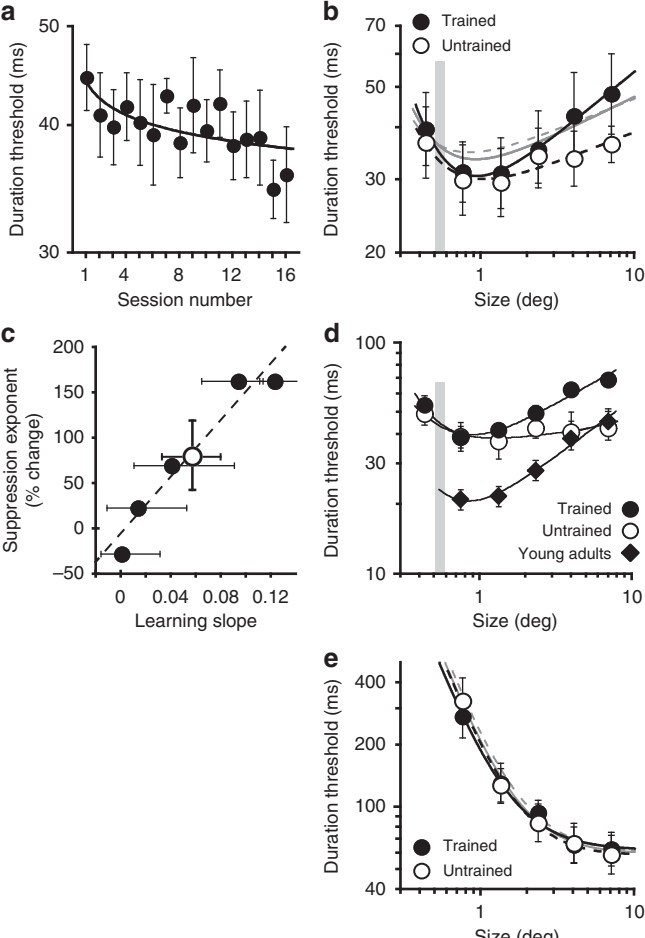

**Fig. 6** Perceptual learning of motion segregation affects spatial suppression. **a** Motion segregation performance as a function of the training session. Data were fit with an exponential function. Error bars are s.e.m. **b** Comparison of pre- and post-training motion direction discriminations at high contrast. Black lines are descriptive model fits to the post-training data (see "Methods" section), with the solid line indicating the trained motion direction. Gray lines are fits to the pre-training data. For clarity, pre-training thresholds are omitted and can be seen in Fig. 5a. Error bars are s.e.m. **c** The relationship between the amount of perceptual learning and the resultant change in spatial suppression strength for the trained motion direction. Error bars are 95% CI, revealing significant learning in 3 observers. The empty circle shows the average overall observers, with s.e. m as error bars. **d** Same as (**b**), except only showing data for observers who exhibited significant perceptual learning. For comparison, data for young adults is replotted from Fig. 5b (averaged over motion direction). Note that the y-axis is larger than in (**b**) to accommodate young adult data. **e** Same as (**b**), but for low contrast stimuli

segregation) further helps to rule out possible confounding effects of attention. Namely, it rules out the possibility that the observed age-related deficits were due to older adults having trouble attending to small moving stimuli. However, it is worth noting that small stimulus thresholds remain abnormally elevated for older adults, and this aspect of our results might be due, at least in part, to older adults' reduced ability to focus spatial attention on small stimuli.

In sum, we found that perceptual training of motion segregation can alleviate age-related impairments in segmenting moving objects from moving backgrounds. Of note, this perceptual learning caused increased thresholds for perception of large, background-like motion but no changes in the sensitivity

**Table 1 Model parameters, comparing young and older adults**

|  |  | $A_e$ | $A_i$ | $\alpha$ | $\beta$ | $\beta/\alpha$ |
|---|---|---|---|---|---|---|
| Young adults | Estimates | 204.2 | 28.6 | 0.57 | 0.76 | 1.34 |
|  | 95% CI | [184.2, 219.3] | [26.4, 30.9] | [0.49, 0.65] | [0.67, 0.87] | [1.32, 1.37] |
| Older adults | Estimates | 122.6 | 18.2 | 0.75 | 0.88 | 1.18 |
|  | 95% CI | [98.4, 147.1] | [15.0, 22.5] | [0.58, 0.97] | [0.66, 1.15] | [1.13, 1.25] |
|  | P-values | <0.001 | <0.001 | 0.026 | 0.19 | <0.001 |

**Table 2 Model parameters, comparing pre-training vs. post-training for the trained motion direction**

|  |  | $A_e$ | $A_i$ | $\alpha$ | $\beta$ | $\beta/\alpha$ |
|---|---|---|---|---|---|---|
| Pre-training | Estimates | 109.5 | 18.7 | 0.80 | 0.96 | 1.21 |
|  | 95% CI | [78.8, 147.9] | [11.4, 28.1] | [0.62, 1.09] | [0.71, 1.30] | [1.15, 1.28] |
| Post-training | Estimates | 117.3 | 21.8 | 0.83 | 1.06 | 1.28 |
|  | 95% CI | [87.9, 138.5] | [15.1, 27.1] | [0.74, 0.97] | [0.92, 1.22] | [1.26, 1.34] |
|  | P-values | 0.34 | 0.26 | 0.39 | 0.22 | 0.020 |

to small, object-like motions, highlighting the crucial role of background insensitivity in motion segregation.

**A mechanistic model of spatial suppression.** In the previous section, data were fit with a descriptive model. We chose that model as it is both very simple, with only three free parameters, and has been shown to closely capture spatial suppression and summation in both young and older adults[47]. However, while that model is appropriate to detect changes in behavioral spatial suppression, it is agnostic about mechanisms underlying spatial suppression. To gain insights into what changes might underlie the observed changes in spatial suppression, we also fit the data with a mechanistic model[54]. The model captures divisive interactions between responses of excitatory center and inhibitory surround regions, with responses varying with both stimulus contrast and size[54]. See Methods for implementation details. This model can capture spatial suppression and summation, as well as account for perceptual differences in special populations, such as older adults[54] and children with autism[55]. Motivated by previous work[54,55], we focused on estimating excitatory and inhibitory spatial pooling sizes ($\alpha$, $\beta$) and response gain parameters ($A_e$, $A_i$), investigating how these parameters change as a function of motion segregation training. Because of the mechanistic nature of the model, we fit low and high contrast data together with the same model. We fit the model to each observer individually as well as to the observers who showed significant perceptual learning of motion segregation.

The mechanistic model provided a good fit to the data (Tables 1−3: all $R^2 > 0.931$, $\chi^2 < 1.27$, $P > 0.99995$). When compared to younger adults (Table 1), older adults exhibited weaker excitatory and inhibitory gains, as well as broader spatial properties that, relative to young adults, were less biased toward inhibitory spatial pooling (i.e., smaller $\beta/\alpha$). This pattern of differences is consistent with prior results for higher spatial frequency stimuli[54]. Considering the effects of training (Tables 2 and 3), modeling results provided statistically significant evidence indicating that motion segregation training leads to an increase in the relative size of the inhibitory spatial pooling ($\beta/\alpha$). This was the case when comparing post-training to pre-training models (Table 2) and when comparing untrained and trained motion directions at post-training (Table 3; Fig. 7). For the latter comparison, we also observed an increase in the inhibitory spatial pooling (Table 3). Considering model fits for individual observers, $\beta/\alpha$ significantly increased for two observers that exhibited the strongest learning (the rightmost points in Fig. 6c),

both when comparing pre- and post-training ($Ps < 0.047$ see "Methods" section for bootstrap analysis details) and when comparing trained and untrained motion directions ($Ps < 0.004$).

**Discussion**
The defining perceptual consequence of spatial suppression is insensitivity to supra-threshold background-like motions. This effect is widely replicated[12] and strong: duration thresholds for motion discriminations of large stimuli can be a factor of two (Fig. 2a) or more[31] times higher than those for perceiving small moving objects. Here, we show that this seemingly maladaptive loss of sensitivity, in fact, is associated with the promotion of an important visual function: motion segregation. Both stimulus-dependent and individual variations in spatial suppression strength predict observers' ability to segregate moving objects. Moreover, perceptual learning of motion segregation caused strengthening of spatial suppression. These behavioral findings support theoretical work[13,16,17] that argues for functional involvement of surround suppression in motion segregation. Neurophysiological studies also postulated that motion segregation is a functional role of surround suppression[15,56]—a hypothesis receiving some support[19]. Using psychophysical spatial suppression as a behavioral correlate of neural surround suppression mechanisms[32–35], we provide the first direct behavioral evidence for this link.

How does spatial suppression facilitate motion segregation? Consider a motion segregation situation involving a small moving object appearing within a background of motion. Because of spatial suppression, the visual sensitivity to background motion will be considerably worse than the visibility of the object's motion. This is particularly true for very brief exposures for which large background motion is imperceptible yet the motion of a small object is readily visible[57]. That is also evident in the present results (Figs 1, 3), where stimulus durations needed for motion segregation are between 1/3rd to 1/7th of durations needed to perceive large background motions. Through this insensitivity to background motion, the visual system effectively accomplishes background subtraction and, as a result, increases visual saliency of object motion[16]. From a computational standpoint, this is an efficient strategy because motion segregation is achieved by suppressing irrelevant information, rather than enhancing relevant figure information, reducing the overall responsiveness of the system[46]. Consistent with this idea, motion segregation learning only affected the sensitivity to large motions, having no effect on the discriminability of small moving stimuli

**Table 3 Model parameters, comparing post-training results for untrained and trained motion directions**

| | | $A_e$ | $A_i$ | $\alpha$ | $\beta$ | $\beta/\alpha$ |
|---|---|---|---|---|---|---|
| Untrained | Estimates | 113.6 | 17.0 | 0.77 | 0.87 | 1.13 |
| | 95% CI | [79.2, 139.2] | [12.9, 20.1] | [0.69, 0.97] | [0.75, 1.03] | [1.07, 1.25] |
| Trained | Estimates | 117.3 | 21.8 | 0.83 | 1.06 | 1.28 |
| | 95% CI | [87.9, 138.5] | [15.1, 27.1] | [0.74, 0.97] | [0.92, 1.22] | [1.26, 1.34] |
| | $P$-values | 0.39 | 0.11 | 0.27 | 0.039 | <0.001 |

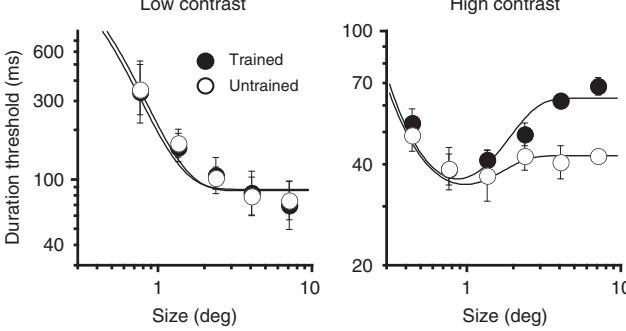

**Fig. 7** Mechanistic model fits. Low and high contrast data are shown for observers that exhibited significant perceptual learning, contrasting the trained and the untrained motion direction. Lines are mechanistic model fits as described in the main text and "Methods" section. All error bars are s.e.m.

(Fig. 6b, d). Moreover, the object detection experiment (Fig. 2) showed that vision can fully account for the moving background under conditions that support spatial suppression. This concept of background subtraction is found in a number of domains, including in the retina where it functions to eliminate uniform image motion due to small eye movements[58]. Our ability to estimate motion of objects during self-motion also appears to depend on a mechanism that globally subtracts optic flow signals caused by self-motion[59]. In computer vision, background subtraction is used to segment foreground objects, a strategy that is particularly effective when there is a common background motion[17,36–39]. Furthermore, background subtraction can be used to enhance odor discrimination[60], suggesting that it may be a general computational strategy used across different modalities.

The key result of our study is the firm establishment of a link between behavioral spatial suppression, a seemingly maladaptive visual phenomenon, with an important functional role: visual motion segregation. In addition, we also applied a model[54,55] to gain insights into what mechanistic changes in surround suppression might occur together with improvements in motion segregation. The model showed that, as a result of motion segregation training, the size of the suppressive spatial pooling becomes larger relative to the integrative (i.e., excitatory) spatial pooling. Such a mechanism would make responses to large stimuli weaker, plausibly leading to stronger behavioral spatial suppression. Notably, the previous work[54] indicates that aging is associated with abnormal spatial pooling of motion that favors spatial integration of motion signals. Our study suggests that motion segregation training may modify this age-related abnormality. These results, while providing clues into potential underlying mechanisms of spatial suppression, should be interpreted with caution. As with all mechanistic models, the one presented here is constrained by assumptions made about underlying mechanisms—a question that is still under debate for surround suppression in motion[34,35,61]. Note, however, that this limitation does not affect the demonstrated link between spatial suppression and motion segregation (Fig. 1–6), as both of these

characteristics of motion perception are operationalized on a behavioral level.

We argue that many of the relevant motion segregation processes occur on relatively brief time scales[4], likely driven by evolutionary pressure to rapidly segregate moving objects. Our results show that a 16 ms stimulus exposure is sufficient to segregate a motion-defined object (Fig. 2b)—a perceptual ability that we link with mechanisms underlying spatial suppression. While spatial suppression effects have been found over a range of time scales[31,57,62–65], marked suppression of large moving stimuli is found only for brief presentations[31,57]. This poor perceptual sensitivity to large moving stimuli when presented briefly is consistent with a strong bias of MT responses toward representing object rather than background motions at brief durations[32]. While surround suppressed MT neurons exhibit good directional selectivity for small brief stimuli, wide-field MT neurons have poor directional selectivity for brief stimuli of any size. In other words, neurons that are well suited to represent background motion do not do so for brief stimuli. Importantly, this study[32] shows that the earliest directionally selective responses in area MT will be those representing motion of small stimuli, effectively segregating those responses from background motion responses. Whether this temporal segregation of responses is explicitly used as a motion segregation cue is an open question.

Thinking broadly, our findings also raise the question of possible negative consequences of suppressing large moving stimuli, since such motions are undoubtedly useful for other visual functions. However, we previously showed that despite its dramatic effects on visibility of large background motions, spatial suppression has little effect on second-order motion[66], motion adaptation[67] and ocular following reflexes (OFR)[68]. This specificity of spatial suppression to perception of first-order motion, we argue, is adaptive. While insensitivity to background motion can improve segregation of moving objects, some motion processes, such as OFR, require high sensitivity to large moving stimuli.

By including older adults in our study, we were able to exploit the additional variability in spatial suppression strength to explore its role in motion segregation. The results revealed that previously reported better-than-normal motion sensitivity to large moving stimuli[42] comes at a cost of impaired motion segregation. Here, we speculate that in the aging visual system, the benefits of rapid motion segregation are outweighed by the need for enhanced spatial integration of motion signals, signals that might be atypically noisy at old age. In other words, the demands of vision at old age shift the discrimination/integration balance toward enhanced integration[69]. Moreover, these results predict that other populations that exhibit abnormally weak spatial suppression (schizophrenia[48], major depression[70]) should also exhibit similar deficits in motion segregation. The observed trade-off between integration and segregation is not unique to motion segregation, but is ubiquitous in visual processing[6,71,72], other sensory modalities[7,8,11] as well as cognitive function[9,10]. In our study, we show that conditions promoting spatial sensitivity and integration, such as low contrast, are associated with poor segmentation, and vice versa (Fig. 1). For older adults, this balance

can be changed as a result of motion segregation training (Fig. 6). While it may seem counterintuitive that perceptual learning of one motion task results in worsening motion perception on another task, this result is entirely consistent with the functional link between spatial suppression and motion segregation. This link appears to be adaptive, where computationally useful suppressive mechanisms are only utilized when motion signals are sufficiently strong. For slow, low contrast and noisy motions, integrative mechanisms dominate[31,43].

In summary, we report behavioral evidence for both correlational and causal links between motion segregation and mechanisms underlying spatial suppression. We conclude that perceptual insensitivity to large, background-like moving stimuli effectively implements background subtraction, in turn enhancing the relative visibility of moving objects.

## Methods

**General methods.** All procedures complied with guidelines set by institutional review boards at Vanderbilt University and University of Rochester. All participants gave written informed consent. Stimuli were created in MATLAB and Psychophysics Toolbox. All experiments measured duration thresholds[31,35,42,46,47,51,57,70,73–75], estimating the minimum stimulus duration required for threshold-level performance. Duration thresholds were estimated by interleaved QUEST staircases converging to 82% correct[76]. Feedback was provided for each trial.

**Covariation of motion segregation and spatial suppression.** In Experiment 1, stimuli were shown on a linearized monitor (1024 × 768 resolution, 120 Hz). Viewing was binocular at 83 cm (yielding 1.5 × 1.5 arcmin per pixel). Ambient and background illuminations were 4.8 cd m$^{-2}$ and 60.5 cd m$^{-2}$. The stimuli were moving random-texture patterns consisting of light and dark elements (each 3 × 3 arcmin) presented in a 2D raised cosine envelope (Fig. 1a, b). From frame to frame of the animation, half of the pixels shifted in one direction (6.2° s$^{-1}$) while the remaining pixels were randomly regenerated (50% correlation). 50% correlation was used to slightly increase task difficulty and, thus, avoid possible floor effects. These conditions produced vivid motion perception at suprathreshold exposure durations. The stimulus size was fixed and relatively large (radius = 5.3°). To allow presentation of brief motion stimuli, stimulus contrast was ramped on and off with a temporal Gaussian envelope (duration is defined as the 2σ of the temporal Gaussian). For each condition, four young adult observers (two females and two males; three naïve observers; age range: 19–25) ran four pairs of interleaved staircases, with the first pair discarded as practice.

The motion discrimination and segregations tasks were conducted at 6 contrast levels each (9–92%). This yielded 12 conditions whose order was randomized. In the motion discrimination task, observers identified motion direction of a random-texture pattern that moved either upward or downward (Fig. 1a). In the motion segregation task, observers discriminated between two motion-defined shapes (Fig. 1b, 45° rightward tilted ellipse vs. 45° leftward tilted ellipse). The texture elements within the elliptic region moved either upward or downward (direction randomly chosen on each trial), while the texture elements outside the elliptic region moved in the opposite direction. Long and short axes of the ellipse were 2.28° and 1.14°, respectively. Note that the single frames from the motion and figure-ground discriminations tasks are indistinguishable (i.e., both are featureless random-texture patterns as shown in Fig. 1a, b).

In Experiment 2, stimuli were presented on a natively linear DLP projector (DepthQ WXGA 360, 1280 × 720 resolution) at 360 Hz with each pixel subtending 2 arcmin of visual angle (146 cm viewing distance). Ambient and background illuminations were 1.8 cd/m$^2$ and 113.7 cd/m$^2$. The same random-texture patterns with light and dark elements each subtending 8′ × 8′ were used as in the other experiments, but they were low-pass filtered (Fig. 2a; cutoff frequency, 1.12 cycles/degree) to eliminate motion blur cues to the object location. Observers' task was to segregate the motion-defined object from its background and report its location (Fig. 2a). The object could appear in one of four quadrants at 2.25° eccentricity, and the background could either move in the opposite direction from the object, or was stationary. The size of the background and the objects were 12° and 0.75° in radius, respectively. Two contrast levels were used: 99% for the high contrast, and 3% for the low contrast condition. Duration thresholds were measured across five different speed levels (1°, 2°, 4°, 8°, and 16° $^{-1}$), with one exception being the exclusion of 1°/s stimuli at low contrast (as such stimuli were difficult to see at any duration). Stimulus duration was defined as full-width at half-height of the temporal contrast envelope, which was a hybrid-Gaussian[33]. Observers completed four blocks (each with two interleaved QUEST staircases per speed level) for both moving and stationary background conditions. The first block was considered as practice and discarded from analysis. Six naïve young adult observers participated in the experiment (five female, one male; age range: 19–28). One observer had difficulty

with 3% contrast stimuli, so this observer did the low contrast task at 3.5%. The data with and without this subject were qualitatively the same.

**Explaining individual differences in motion segregation.** In Experiment 3, stimuli were shown on a linearized high-speed PROCALIX monitor (Totoku, Irving, TX, 21 in, 800 × 600 resolution, 200 Hz). 200 Hz frame rate ensured that our measurements were not affected by frame-sampling limitations[64], allowing us to use fully coherent motion. Viewing was binocular at 114 cm. Ambient and background illuminations were 0.15 and 41.5 cd m$^{-2}$. Stimulus duration was defined in the same manner as in Experiment 2. All stimuli were composed of random-texture patterns that consisted of light and dark elements (each 4.5′ × 4.5′, Fig. 1). To produce motion, texture elements uniformly shifted in position on each frame (i.e., 100% coherence), yielding speed of 5° s$^{-1}$. Stimuli were shown in stationary raised cosine spatial envelopes, whose radii defined stimulus size.

21 young (mean age = 22.5, std. = 4.18, age range = 18–35) and 21 older adults (mean age 67.7, std. = 5.88, age range: 60–84; 13 female, 8 male) participated. Older adults underwent basic cognitive screening (Modified Telephone Interview for Cognitive Status score over 30; absence of diagnosis of Alzheimer's disease, no reported neuropsychological deficits, recent severe medical conditions (e.g., stroke) and severe sensory or motor loss). Four types of moving stimuli were used (Fig. 3a, b), all presented at 99% peak stimulus contrast. In two of the tasks (Fig. 3a), observers discriminated motion direction (up vs. down) of small (r = 0.75°) and large (r = 5°) moving stimuli. The small stimulus size was chosen to be near optimal, while the large stimulus was sufficiently large to yield strong spatial suppression[51]. To estimate motion segregation ability, observers identified the orientation of a motion-defined shape (Fig. 3b, left; 45° left vs. 45° right tilt, randomly chosen) embedded in a large moving background (r = 5°). Long and short radii of the elliptical shape were 0.7° and 0.35°. The pixels within the elliptical figure moved either up or down (randomly chosen), while the pixels outside the elliptical region (i.e., background) moved in the opposite direction. Finally, observers conducted a figure-motion discrimination task. Here, the above-described motion segregation stimulus was used but without background motion. The observers' task was to simply identify its motion direction. Shape orientation was chosen randomly for each block of trials, remaining constant for its duration. Note that all motion discrimination tasks involved discriminating between up and down motions. This was done to avoid response the possible confusion in the motion segregation task (which involved right vs. left tilt decisions).

For each of the four conditions, observers completed four pairs of interleaved staircases conducted in four sessions of trials. Within each session, the order of four conditions was randomized, each tested in a separate block. Before each block, a screen displaying the task instructions and an exaggerated stimulus example was shown. Overall, this procedure yielded four pairs of thresholds for each condition. The first pair was used as task practice and to set the starting points for subsequent staircases. Next, the lowest and the highest results were eliminated, and the remaining thresholds were averaged. We routinely use this exclusion of extreme values when working with subjects that are inexperienced with psychophysics[46,75,77]. For the present data sets (Figs. 3 and 4), removing extreme values improved the average split-half reliability from 0.87 to 0.95 (just looking at older observers, the reliability was 0.94). Data for 1 young and 3 older adults were excluded because data variability (defined as standard deviation, SD, of threshold measurements for individual conditions) was 3 SD higher than the average variability.

**Causal linking of motion segregation and spatial suppression.** Experiment 4 included three parts: pre-training measures, motion segregation training, and post-training measures. The equipment and most stimulus parameters were same as in Experiment 3. Observers were older adults (n = 5; mean age = 68.2, std. = 4.92, age range = 64–76; two females, three males). For both pre- and post-training measures, we characterized size-dependency of motion direction discriminations for two contrasts (7% and 99%). At high contrast, stimulus sizes were r = 0.43°, 0.75°, 1.33°, 2.33°, 4°, and 7°. At low contrast, the smallest size was omitted because of its poor visibility. These measurements were done for both horizontal (left vs. right) and vertical (up vs. down) motion direction discriminations, tested in two consecutive counterbalanced sessions. Each session consisted of 11 randomized blocks of trials, each consisting of one size-contrast combination. In each block, two thresholds were estimated. This entire sequence was repeated on four separate days, yielding a total of 8 thresholds per condition. The first day was taken as practice.

During motion segregation training, each observer completed 16 perceptual training sessions conducted on separate days (taking 4–5 weeks). In each session, observers completed 10–12 blocks of motion segregation, with two interleaved QUEST staircases per block. Motion segregation task was as described above, except that figure and background motions were either both horizontal or both vertical. This was pseudo-randomly assigned for each observer (3 vertical, 2 horizontal), remaining constant throughout the training.

To describe size-dependency of motion perception, we used a simple descriptive model that has been shown to closely capture size and contrast-dependency of duration thresholds for both young and older adults[47]. Namely, we fit the pre- and

post-training results with:

$$f(x) = p_1 x^{k_1} + p_2 x^{k_2}, \text{ with } k_1 \text{ fixed at } -2. \quad (1)$$

where $x$ is the stimulus size, $p_1$ and $p_2$ set the height of the descending (i.e., summation) and ascending (i.e., suppression) portions of the curve, while the $k_1$ and $k_2$ parameters determine the descending and ascending slopes. Parameter $k_1$ was set to $-2$, to incorporate linear temporal summation[47]. We confirmed this assumption by separately fitting $k_1$ as a free parameter, which resulted with the average fitted value across all conditions of $-1.99$. For our purposes, the key parameter is the suppression slope ($k_2$). Change in $k_2$ (namely smaller $k_2$) closely describes age-related changes in spatial suppression[47].

**A mechanistic model of spatial suppression.** Data from the perceptual training experiment (Experiment 4) were fitted with a mechanistic model that captures size- and contrast-dependent interactions between the receptive field center and surround[54]. Namely, the model includes an excitatory center ($E$) and an inhibitory surround ($I$):

$$E(w) = 1 - e^{-\frac{(\frac{w}{\alpha})^2}{2}}, \quad (2)$$

$$I(w) = 1 - e^{-\frac{(\frac{w}{\beta})^2}{2}}, \quad (3)$$

where $w$ is stimulus size (in degrees), while $\alpha$ and $\beta$ are size constants for the center and the surround spatial pooling, respectively. Both the center and surround responses depend non-linearly on stimulus contrast, as determined by the Naka-Rushton function:

$$K_e(c) = A_e \frac{c^{n_e}}{c^{n_e} + c_{50_e}^{n_e}}, \quad (4)$$

$$K_i(c) = A_i \frac{c^{n_i}}{c^{n_i} + c_{50_i}^{n_i}}, \quad (5)$$

where $c$ is stimulus contrast, while excitatory and inhibitory regions are captured by parameters determining response gain ($A_e$, $A_i$), semi-saturation point ($c_{50_e}$, $c_{50_i}$) and slope ($n_e$, $n_i$), respectively. The overall response is determined by the ratio of the responses in the center and surround (i.e., inhibition is divisive):

$$R(w, c) = \frac{K_e(c) \cdot E(w)}{1 + K_i(c) \cdot I(w)}, \quad (6)$$

which is then converted to thresholds:

$$T = \frac{Criterion}{R_0 + R}. \quad (7)$$

*Criterion* and $R_0$ scale the model responses to perceptual thresholds, and were fixed to 20 and 6, respectively, following the previous work[54].

Both at pre-training and at post-training, we estimated spatial pooling size ($\alpha$, $\beta$) and response gain parameters ($A_e$, $A_i$) (the key parameters of interest), while fixing other parameters ($c_{50_e}$, $c_{50_i}$, $n_e$, $n_i$ were fixed to 0.2, 0.2, 3, and 5, respectively). Changing values of fixed parameters (i.e., slopes and semi-saturation constants) had minimal effects on the model fits (largely because the fitted data consisted of only low and high contrasts). The model was fitted using the least squares procedure.

**Statistics.** For most analysis, conventional statistical tests were used (ANOVA, t-test and Pearson correlation). All tests were two-tailed. For models, non-parametric bootstrap was used to obtain 95% confidence intervals (CI) and associated statistical significance for the descriptive model. Specifically, for each observer, individual thresholds (for the condition of interest) were sampled with replacement (9999 and 1000 bootstrap samples for the descriptive and the mechanistic model, respectively), and the model fitted to each set of samples. The 95% confidence intervals and p-values were extracted from the resampled distributions. We compiled difference distributions (pre-training vs. post-training, untrained vs. trained directions (post-training), or younger vs. older (pre-training), depending on the analysis) for each model parameter. p-values were determined by the proportion of samples that "crossed" zero.

**Reporting summary.** Further information on research design is available in the Nature Research Reporting Summary linked to this article.

## Data availability
The data sets generated during and/or analyzed during the current study (Figs. 1–7) are available from the corresponding author on request.

## Code availability
Modeling computer code (MATLAB) is also available on request.

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

## Acknowledgements

This work was supported by a NIH NEI grant R01 EY019295 to D.T and awards NEI P30 EY001319, T32 EY007125 and T32 EY007135-19.

## Author contributions

Conceptualization: D.T., W.J.P., J.S.L. and R.B.; Methodology: D.T. and W.J.P.; Software: D.T. and W.J.P.; Formal analysis: D.T. and W.J.P.; Investigation: D.T., W.J.P., K.C.D. and M.D.M.; Writing—Original draft: D.T.; Writing—Review & Editing: D.T., W.J.P., K.C.D., M.D.M., J.S.L. and R.B.; Funding acquisition: D.T.

## Additional information

**Competing interests:** The authors declare no competing interests.

