## [Transparent Peer Review File · Nature Communications]

Reviewers' comments:

Reviewer #1 (Remarks to the Author):

In this paper, Tadin et al present a series of psychophysical experiments designed to investigate the mechanisms underlying motion perception and motion segregation. They report a strong relation between motion sensitivity and segregation capacity: the more detectable large background-like motion signals are, the harder it is to segregate small motion-defined objects from these backgrounds. They hypothesize that this link is a consequence of suppressive center-surround mechanisms underlying perceptual capabilities in both tasks.

While I find the central claim of this paper interesting and plausible, I found it a tough read and wonder whether it might be more suited for a specialty journal. The logic underlying the different experiments is not easy to grasp, and the evidence for the overarching hypothesis – suppressive center-surround mechanisms get invoked to different degrees by the different manipulations – is always indirect. For example, lowering stimulus contrast does indeed reduce suppressive drive. But it also reduces excitatory drive, so it is not that obvious that the effect of this manipulation necessarily has to be understood as reduced suppression.

To make this paper accessible for a broad audience, and to provide a more direct test of the central hypothesis, I propose to both drastically re-organize the paper, and to do more with the modeling component. Of all the experiments reported, the last one (Figure 6) is the most compelling. I think this experiment is beautiful and should lead the paper. The stimuli, task, and analysis of behavioral reports should be explained better and more extensively in the text to make a general audience appreciate what exactly is being shown here. I would follow this set of empirical observations with a section on a computational model that instantiates the effects of the hypothesized mechanisms. In line with most observer models, I think this model should have an encoding stage and a decoding stage (the model currently used seems more descriptive than mechanistic). The authors can use this model to clearly explain what exactly the effects of their stimulus manipulations and of perceptual learning are on the different components of the model. They could illustrate under what kind of parameter-regime the model can recapitulate the empirical observations, and when it fails to do so. This will provide a more direct test of their hypothesis, and would elevate the importance of this study. It will also allow readers to develop a better intuition for the mechanisms underlying the reported perceptual behavior. The other experiments could (in part) move to supplementary information.

Reviewer #2 (Remarks to the Author):

This is an interesting and solid study that investigated the functional link between motion-based segregation and size-dependency in motion direction sensitivity. The key finding is that reduced directional sensitivity to large moving stimulus is linked to better performance in motion-based segregation, as seen in correlational as well as 'causal' evidence from perceptual learning. While I think the study was well motivated and the results are very informative, there are a number of issues that the authors should consider and discuss/clarify.

1. The title is too broad. There are many other aspects of motion-based figure-ground segmentation that are not addressed in this paper. Need a more pertinent title.
2. How much of the results reported in Fig 1 could be attributed to the 'raised cosine envelope'? With such a soft window, different contrast levels would be associated with different 'effective' visible area of the stimulus, i.e., at low contrast, the stimulus effectively became smaller. Would the same results hold with a hard window?

3. The authors often describe the large 5 deg moving stimuli as 'background motion' even when they were the 'target' for the motion discrimination. Occasionally they were called 'background-like'. I understand that in the segregation task, the surrounding area outside the central target shape should be background; but in the motion discrimination task, the whole stimulus was the target. What makes a stimulus 'background-like'?
4. The authors should spell out what they meant by 'suppression'. What is suppressed? Is it the center suppressed by surround, or some suppressive mechanism applied to the surround region, or simply the (center+ surround-) receptive field structure?
5. Could the results be more straightforwardly interpreted as the consequences of distribution of receptive field sizes of motion-sensitive neurons? Perceptual training could selectively increase the sensitivity of a sub-population of neurons with certain receptive field size and decrease the sensitivity of neurons with larger (or smaller) receptive field sizes?
6. Does the ability (and the flexibility) to focus spatial attention play a role, both in the difference of size-dependency observed between the young and older subjects, and in the consequence of motion-segregation training?
Looking at Fig 3C/3D, one could argue that the older and young subjects performed about equally in motion discrimination for the large stimuli. Older subjects were not as good as young subjects when dealing with small stimuli, as in motion discrimination for the small stimuli, and identify the small motion-defined shape. Could this reflect a reduced ability in older subjects to focus spatial attention?

Reviewer #3 (Remarks to the Author):

I apologize for taking so long to submit my review.

Synopsis: The submitted manuscript describes several experiments that examine the link between spatial suppression and motion segregation. Spatial suppression refers to a psychophysical phenomenon in which motion discrimination often deteriorates for large, high-contrast stimuli. For example, the stimulus duration needed to discriminate leftward and rightward drifting high contrast sine wave gratings increases significantly as stimulus size increases. Spatial suppression, which has been linked to surround suppression in some MT cells, has been hypothesized to play a role in figure-ground organization, especially the detection of moving objects presented against moving backgrounds. The authors examine this hypothesis in several psychophysical experiments that measure motion direction discrimination and motion segregation thresholds in conditions in which spatial suppression is strong or weak. The results are quite clear: motion segregation was best in conditions in which spatial suppression is strong. For example, In Experiment 1, conditions that use large, high-contrast stimuli yielded high direction discrimination thresholds (consistent with spatial suppression) but low motion segregation thresholds. Experiment 2 showed that, at high contrast, detecting a moving textured target was >>easier<< when the target was presented against a large moving background than when it was presented against a large static background. And Experiment 3 found that motion segregation is impaired in older adults (who have weak spatial suppression) compared to young adults. All of these results are consistent with the idea that spatial suppression and motion segregation are associated with each other. Experiment 4 examines whether there is a causal link between the two phenomena by using perceptual learning (PL) to improve the performance of older adults in a motion segregation task. If practice improves motion segregation, and if motion segregation depends on spatial suppression, then PL should produce changes in spatial suppression. This is a very interesting prediction because the effects of PL often are limited to the particular stimuli and tasks experienced during practice. The results are clear: practice did improve motion segregation in amounts that varied across individuals, and these improvements were correlated with changes in spatial suppression (e.g., Figure 6C). Notably, the largest improvements in motion segregation were associated with an >>increase<<

in motion discrimination thresholds for large high-contrast patterns.

General Comments: The submitted manuscript investigates very interesting issues in visual perception. Although many investigators have proposed a link between spatial/surround suppression and motion segregation, I do not know of any demonstrations of that link that are as clear and convincing as those presented in the submitted paper. The results obtained with older observers are very clear and the results of the learning experiment are amazing. The paper is well written and easy to follow, and I think that the results are very exciting. Finally, I believe that the methods contain sufficient detail for other researchers to replicate and extend the experiments. In short, this is one of the best papers that I've read in a long time, and my recommendation would be to accept the manuscript for publication.

Reviewer #1: *In this paper, Tadin et al present a series of psychophysical experiments designed to investigate the mechanisms underlying motion perception and motion segregation. They report a strong relation between motion sensitivity and segregation capacity: the more detectable large background-like motion signals are, the harder it is to segregate small motion-defined objects from these backgrounds. They hypothesize that this link is a consequence of suppressive center-surround mechanisms underlying perceptual capabilities in both tasks.*

While I find the central claim of this paper interesting and plausible, I found it a tough read and wonder whether it might be more suited for a specialty journal. The logic underlying the different experiments is not easy to grasp, and the evidence for the overarching hypothesis – suppressive center-surround mechanisms get invoked to different degrees by the different manipulations – is always indirect. For example, lowering stimulus contrast does indeed reduce suppressive drive. But it also reduces excitatory drive, so it is not that obvious that the effect of this manipulation necessarily has to be understood as reduced suppression. To make this paper accessible for a broad audience, and to provide a more direct test of the central hypothesis, I propose to both drastically re-organize the paper, and to do more with the modeling component. Of all the experiments reported, the last one (Figure 6) is the most compelling. I think this experiment is beautiful and should lead the paper.

We thank the reviewer for these comments – and wish to express our agreement: Indeed, the links shown by experiments in our study start as indirect (and correlational) and gradually culminate with the last (Figure 6) experiment which shows a direct link between spatial suppression and motion segregation (we agree with the reviewer’s choice of adjective, but, unfortunately, we can’t use “beautiful” in the paper). We selected this order as early experiments generate hypotheses that are then tested by subsequent experiments. During the revision process, we tried the organization suggested by the reviewer, but we felt that it did not work as well as the “build-up” organization. To address the reviewer’s concern, we now provide an explicit outline of the “build-up” organization at the beginning of the Results, and suggest that readers, if they wish, can move to Figure 6 section (3rd section) for the most compelling link between spatial suppression and motion segregation, treating the first two sections similar to supplementary materials. We also edited the beginning of that section so that it does not require a reading of previous sections. This is perhaps unorthodox, but it does afford flexibility to the reader as to how to approach the results.

The stimuli, task, and analysis of behavioral reports should be explained better and more extensively in the text to make a general audience appreciate what exactly is being shown here.

We now provide additional information about the tasks and analysis, expanding both the main text and relevant figure captions.

I would follow this set of empirical observations with a section on a computational model that instantiates the effects of the hypothesized mechanisms. In line with most observer models, I think this model should have an encoding stage and a decoding stage (the model currently used seems more descriptive than mechanistic). The authors can use this model to clearly explain what exactly the effects of their stimulus manipulations and of perceptual learning are on the different components of the model. They could illustrate under what kind of parameter-regime the model can recapitulate the empirical observations, and when it fails to do so. This will provide a more direct test of their hypothesis, and would elevate the importance of this study. It will also allow readers to develop a better intuition for the mechanisms underlying the reported perceptual behavior. The other experiments could (in part) move to supplementary information.

Indeed, the model used in the previous version of the manuscript is only descriptive. We now also include a mechanistic model. We selected the original model as it is a simple (only 3 free parameters) and an

established way to capture spatial suppression, allowing us to test whether spatial suppression changes as a result of motion segregation training. For these reasons, we're still keeping the descriptive model, but now follow it with a model that aims to answer what mechanistic changes might underlie the observed change in spatial suppression (i.e., the change caused by improvements in motion segregation).

The results suggest that, as a result of motion segregation training, the size of the suppressive spatial pooling becomes larger relative to the integrative (i.e., excitatory) spatial pooling. Notably, previous work (Betts et al., 2012, Vision Research) indicates that aging is associated with abnormal spatial pooling of motion signals that favors integration of signals. Our study suggests that motion segregation training can modify this age-related abnormality.

We thank the reviewer for suggesting an addition of a mechanistic model. Our initial hesitation for doing that was that such models require larger numbers of parameters (in the revision, we're using previous work to restrict parameter numbers). As noted above, we are keeping the descriptive model because it helps demonstrate that behavioral spatial suppression, a seemingly maladaptive visual phenomenon, is associated with an important functional role: motion segregation. The added model now provides details about mechanisms that might underlie observed behavior.

Reviewer #2: *This is an interesting and solid study that investigated the functional link between motion-based segregation and size-dependency in motion direction sensitivity. The key finding is that reduced directional sensitivity to large moving stimulus is linked to better performance in motion-based segregation, as seen in correlational as well as 'causal' evidence from perceptual learning. While I think the study was well motivated and the results are very informative, there are a number of issues that the authors should consider and discuss/clarify.*

1. The title is too broad. There are many other aspects of motion-based figure-ground segmentation that are not addressed in this paper. Need a more pertinent title.

As suggested, we changed the title to a more specific one: "Mechanisms of motion-based figure-ground segmentation: the role of spatial suppression."

2. How much of the results reported in Fig 1 could be attributed to the 'raised cosine envelope'? With such a soft window, different contrast levels would be associated with different 'effective' visible area of the stimulus, i.e., at low contrast, the stimulus effectively became smaller. Would the same results hold with a hard window?

The reviewer is correct: changing contrast will also change the effective stimulus sizes, which will also affect spatial suppression strength. This is now highlighted in the manuscript. This does not, however, affect the conclusion of Exp. 1 for two reasons:

First, the effect (on thresholds) of this size change will be less than that of changing contrast. If we conservatively define the visibility cut-off at 2% contrast, then the radius of the lowest contrast stimulus will be 23.9% smaller than that of the highest contrast stimulus. While notable, such a size change would not have a large effect on thresholds (e.g., examine Figure 5B)

More importantly, this effective size change, like changes in contrast, affects both tasks equally (motion discrimination and motion segregation), so it does not change the main conclusion of this experiment – that changes in spatial suppression are correlated with changes in motion segregation. We now emphasize that the observed change in spatial suppression is due to both changes in contrast and contrast-dependent changes in stimulus size (also as shown in prior studies; Tadin et al., 2003).

[We have done experiments with hard windows in the past, but subjects can easily "cheat" on such stimuli by simply monitoring one edge of the stimulus and looking to see if elements either appear or disappear. Moreover, that clue is hard to ignore once noticed.]

3. The authors often describe the large 5 deg moving stimuli as 'background motion' even when they were the 'target' for the motion discrimination. Occasionally they were called 'background-like'. I understand that in the segregation task, the surrounding area outside the central target shape should be background; but in the motion discrimination task, the whole stimulus was the target. What makes a stimulus 'background-like'?

Thanks for this comment. We made two changes in response. First, we re-organized Exp 1. Now we lead with the motion segregation task (where the stimulus is a moving object on a moving background), and then clearly state that the goal of the discrimination task is to estimate the visibility of the background motion presented on its own. This allowed us to eliminate the need to use “background-like” in Exp 1. More broadly, the reason we used the term “background-like” is that our argument (i.e., the main hypothesis of the paper) that the large, high contrast stimuli are treated as backgrounds – even if they are target stimuli. This is consistent with classic observations by Rubin and others (Künnapas, 1957; Oyama, 1960) that smaller and larger stimuli are more likely to be perceived as figures and backgrounds, respectively (work now mentioned in the MS). Our experiments are consistent with this classic view. We now reference this classic work. Moreover, we also removed the term “background-like” early in the paper, and only start using it as we present more evidence for our main hypothesis.

4. The authors should spell out what they meant by 'suppression'. What is suppressed? Is it the center suppressed by surround, or some suppressive mechanism applied to the surround region, or simply the (center+ surround-) receptive field structure?

The reviewer’s comments highlight a need for better definitions on our side. We now clearly differentiate “spatial suppression” (which is defined behaviorally as poor motion discriminations of large, high contrast stimuli) from underlying mechanisms that might cause spatial suppression. For the latter, we use the term “surround suppression.” A new mechanistic model that is now included in the manuscript provides more specificity on what computations might underlie spatial suppression (and training-induced changes in spatial suppression).

5. Could the results be more straightforwardly interpreted as the consequences of distribution of receptive field sizes of motion-sensitive neurons? Perceptual training could selectively increase the sensitivity of a sub-population of neurons with certain receptive field size and decrease the sensitivity of neurons with larger (or smaller) receptive field sizes?

The reviewer’s intuition is consistent with the results from the mechanistic model added to the paper. The modeling suggests that data could be explained by training-induced changes to the relative extent of excitatory and inhibitory spatial pooling – in a way that it would make responses to large stimuli weaker (i.e., result in weaker behavioral spatial suppression). We, however, are being cautious not to overinterpret those results, but see value in providing suggestions on what might be happening at a mechanistic level.

6. Does the ability (and the flexibility) to focus spatial attention play a role, both in the difference of size-dependency observed between the young and older subjects, and in the consequence of motion-segregation training? Looking at Fig 3C/3D, one could argue that the older and young subjects performed about equally in motion discrimination for the large stimuli. Older subjects were not as good as young subjects when dealing with small stimuli, as in motion discrimination for the small stimuli, and identify the small motion-defined shape. Could this reflect a reduced ability in older subjects to focus spatial attention?

We now include discussion of possible attentional issues. There are two points that help us rule out the effects of attention. For low contrast stimuli, older adults have considerably higher thresholds for all

stimulus sizes (Figure 5). This replicates results from Betts et al. (2005, Neuron, their Figure 2A). Here, if attention were a factor, then we would expect at least equal changes at low contrast. Similarly, we find no training induced changes at low contrast (Figure 6E) and for small, high-contrast moving stimuli (Figure 6B,D; gray bar). That is, training did not affect the ability to perceive small moving stimuli of any contrast.

Related to this issue (but not something that is appropriate to put in this paper), one of the authors (KD) spent first two years of his PhD trying to find a way to modulate spatial suppression with attention. To our surprise at the time, we consistently got null results. So, it appears that spatial suppression is relatively immune to different types of attentional modulation.

Reviewer #3: *Synopsis: The submitted manuscript describes several experiments that examine the link between spatial suppression and motion segregation. Spatial suppression refers to a psychophysical phenomenon in which motion discrimination often deteriorates for large, high-contrast stimuli. For example, the stimulus duration needed to discriminate leftward and rightward drifting high contrast sine wave gratings increases significantly as stimulus size increases. Spatial suppression, which has been linked to surround suppression in some MT cells, has been hypothesized to play a role in figure-ground organization, especially the detection of moving objects presented against moving backgrounds. The authors examine this hypothesis in several psychophysical experiments that measure motion direction discrimination and motion segregation thresholds in conditions in which spatial suppression is strong or weak. The results are quite clear: motion segregation was best in conditions in which spatial suppression is strong. For example, In Experiment 1, conditions that use large, high-contrast stimuli yielded high direction discrimination thresholds (consistent with spatial suppression) but low motion segregation thresholds. Experiment 2 showed that, at high contrast, detecting a moving textured target was >>easier<< when the target was presented against a large moving background than when it was presented against a large static background. And Experiment 3 found that motion segregation is impaired in older adults (who have weak spatial suppression) compared to young adults. All of these results are consistent with the idea that spatial suppression and motion segregation are associated with each other. Experiment 4 examines whether there is a causal link between the two phenomena by using perceptual learning (PL) to improve the performance of older adults in a motion segregation task. If practice improves motion segregation, and if motion segregation depends on spatial suppression, then PL should produce changes in spatial suppression. This is a very interesting prediction because the effects of PL often are limited to the particular stimuli and tasks experienced during practice. The results are clear: practice did improve motion segregation in amounts that varied across individuals, and these improvements were correlated with changes in spatial suppression (e.g., Figure 6C). Notably, the largest improvements in motion segregation were associated with an >>increase<< in motion discrimination thresholds for large high-contrast patterns.*

General Comments: The submitted manuscript investigates very interesting issues in visual perception. Although many investigators have proposed a link between spatial/surround suppression and motion segregation, I do not know of any demonstrations of that link that are as clear and convincing as those presented in the submitted paper. The results obtained with older observers are very clear and the results of the learning experiment are amazing. The paper is well written and easy to follow, and I think that the results are very exciting. Finally, I believe that the methods contain sufficient detail for other researchers to replicate and extend the experiments. In short, this is one of the best papers that I've read in a long time, and my recommendation would be to accept the manuscript for publication.

We thank the reviewer for these comments.

REVIEWERS' COMMENTS:

Reviewer #1 (Remarks to the Author):

The authors addressed my concerns. The clarity of the text has improved, and the inclusion of the mechanistic model offers an explicit hypothesis for the nature of the mechanism underlying the observed effects. It is nice work.

Reviewer #2 (Remarks to the Author):

The revised manuscript has addressed my concerns about the correlated changes between contrast and effective window size, and about the potential role of spatial attention in the training effect. However I don't think the results 'ruled out' (the term used by the authors) the potential attention effect though, since there could be interaction between the stimulus size (in the case of low contrast, effective stimulus size) and attention. The fact that there was not observable effect with attentional manipulation in young adults does not mean that the training effect in older adults could not be mediated by attention.

That said, I agree that the collective evidence is strong in support of a link between spatial suppression and motion segregation.

REVIEWERS' COMMENTS:

Reviewer #1 (Remarks to the Author):

The authors addressed my concerns. The clarity of the text has improved, and the inclusion of the mechanistic model offers an explicit hypothesis for the nature of the mechanism underlying the observed effects. It is nice work.

We thank the reviewer for these comments.

Reviewer #2 (Remarks to the Author):

The revised manuscript has addressed my concerns about the correlated changes between contrast and effective window size, and about the potential role of spatial attention in the training effect. However I don't think the results 'ruled out' (the term used by the authors) the potential attention effect though, since there could be interaction between the stimulus size (in the case of low contrast, effective stimulus size) and attention. The fact that there was not observable effect with attentional manipulation in young adults does not mean that the training effect in older adults could not be mediated by attention.

That said, I agree that the collective evidence is strong in support of a link between spatial suppression and motion segregation.

We thank the reviewer for these comments, and we agree that we cannot completely rule out a role of attention. The manuscript now includes this text:

“However, it is worth noting that small stimulus thresholds remain abnormally elevated for older adults, and this aspect of our results might be due, at least in part, to older adults’ reduced ability to focus spatial attention on small stimuli.”